# A mouse model of early sporadic tau pathology induces neurogenic plasticity in the hippocampus

Alyssa M. Ash[1], Sue Rim Baek[1], Peyton Holder[1], Sara Singh[1], Nicholas P. Vyleta[1], Matthew Willman[2], Kevin R. Nash[2], Jason S. Snyder [1]*

**1** Department of Psychology, Djavad Mowafaghian Center for Brain Health, University of British Columbia, Vancouver, British Columbia, Canada, **2** Department of Molecular Pharmacology and Physiology, Morsani College of Medicine, University of South Florida, Tampa, Florida, United States of America

* jasonsnyder@psych.ubc.ca

## Abstract

Alzheimer's disease varies by sex but is broadly characterized by widespread neurodegeneration and the accumulation of insoluble amyloid plaques and neurofibrillary tangles. However, at the earliest stages of the disease cell death and pathological tau are localized to the entorhinal cortex. In particular, the lateral entorhinal cortex, and its functions in object-related memory, are among the most vulnerable in aging. Notably, the entorhinal cortex projects directly to the dentate gyrus subregion of the hippocampus, where neurogenesis proceeds throughout adult life. Immature, adult-born neurons provide plasticity at the entorhinal-dentate pathway and they may be uniquely responsive, or vulnerable, to early entorhinal tau pathology. To test this, we injected a human tau-expressing recombinant adeno-associated virus (hTau) into the lateral entorhinal cortex and used male and female AsclCreER mice to birthdate downstream dentate neurons born in early postnatal development or adulthood. Consistent with known roles in neurodegeneration, lateral entorhinal hTau expression caused a loss of mushroom spines in downstream dentate gyrus neurons of male and female mice and reduced dendritic complexity of adult-born neurons in male mice. Presynaptic hTau also increased neurogenesis levels and increased the density of thin spines on adult-born neurons in both male and female mice. Consistent with spine addition, hTau increased the slope of the synaptic input-output curves; amongst adult-born neurons, this was due to a specific effect on synapses in male mice. hTau did not alter the magnitude of long-term potentiation at entorhinal synapses onto adult- or developmentally-born neurons. Thus, in a novel model of early sporadic tau pathology, there are changes consistent with neurodegeneration but also compensatory neuroplastic changes, caused in part by neurogenesis. Since immature neurons have also been identified in the human dentate gyrus, a similar neurogenic plasticity may help maintain entorhinal-hippocampal formation in pathological aging.

**Data availability statement:** All relevant data are within the manuscript and its Supporting Information files.

**Funding:** JSS received funding from the Canadian Institutes of Health Research. The funders did not play any role in the study design, data collection and analysis, decision to publish, or preparation of the manuscript.

**Competing interests:** The authors have declared that no competing interests exist.

## Introduction

As baby boomers age, Alzheimer's disease (AD) is predicted to increase in prevalence such that over 1% of the world's population may soon have the disease [1]. Unfortunately, AD pathology spreads irreversibly [2] and neuropathologic changes are often present at the earliest stages of the disease continuum [3]. Despite these stark statistics, offsetting disease progression for even a short duration can have a meaningful impact by reducing the amount of time patients are afflicted [1,4,5]. It is critical, therefore, that the earliest stages of the disease are better understood so that treatments that offset AD progression can be developed.

Alzheimer's disease is more prevalent in females [6,7] and is traditionally associated with widespread accumulation of insoluble Aβ plaques and hyperphosphorylated tau aggregates (tangles). However, in early stages of sporadic AD and human aging, tau pathology is found primarily in the entorhinal cortex (EC) before spreading, possibly through synaptic pathways, to the rest of the brain [8–11]. Human imaging data suggests that female vulnerability may be explained, at least in part, by elevated tau in the EC and greater rate of tau spreading throughout the brain [12–16]. The EC is the primary source of synaptic input to the hippocampus, providing high-dimensional sensory information for memory encoding [17]. It is specifically layer II neurons, the subpopulation that projects to the dentate gyrus (DG) subregion of the hippocampus, that appear to be the most vulnerable: both mild cognitive impairment and the earliest stages of AD are associated with > 50% loss of layer II EC neurons [18,19] and high-resolution imaging reveals specific age-related thinning of this pathway in humans [20], which is associated with impairments in hippocampal memory for the fine details of experience [21,22]. Alzheimer's Disease has classically been viewed as disease of the synapse [23,24]. Indeed, synapse loss in the EC-DG pathway has been touted as the most reliable biological correlate of dementia [25], where memory deficits arise due to a hippocampus that is functionally disconnected from the neocortex [26]. Identifying sources of synaptic plasticity in the EC-DG pathway therefore have great potential to stave off the progression of pathology.

The EC is not homogeneous but consists of medial (MEC) and lateral (LEC) subdivisions that convey information about the context and content of experience, respectively [27,28]. Recent work has identified an anterolateral portion of the human EC that is homologous to the rodent LEC [29,30], accumulates tau early in aging [9] and may serve as a source for spreading tau throughout the brain [31]. Behavioral studies in humans also point to earlier decline of the LEC, which is linked to specific initial disruption of LEC functions in object-related memory [32–35]. Age-related vulnerability in the LEC-DG pathway may be conserved, as LTP at this pathway declines as early as 8 months of age in rodents (i.e., early middle age) [36].

There is also heterogeneity in the primary target of the EC due to ongoing neurogenesis and the presence of immature neurons in the DG [37]. It is typically appreciated that adult-born neurons pass through an immature critical period from ~3–6 weeks of age when they have enhanced afferent [38–40] and efferent [41] synaptic plasticity and greater intrinsic excitability [42]. The question arises as to whether mature vs immature neurons may be differentially impacted

by age-related presynaptic tau pathology. Intriguing support for a regenerative function comes from evidence that adult-born neuron dendrites and synaptic structures develop over ~25% of the lifespan (in rodents) [43], they preferentially connect with the LEC [44,45], and they acquire the capacity for long-term synaptic plasticity at their LEC inputs over an extended window of several months [46].

Transgenic mice expressing familial amyloid-related mutations have been the predominant model for studying age-related dementia, often revealing impaired EC-DG synaptic plasticity, DG neuronal atrophy and impairments in learning and memory [47–49]. More recent models have incorporated tau and reveal damage in the EC-DG circuit [50,51]. However, a limitation of most existing models is that tau expression does not recapitulate the pattern that is observed in sporadic human aging and AD pathology, where abnormal tau originates in the EC. To begin to address this issue the neuropsin promoter [52] has been used in an attempt to restrict tau expression to the EC. While these mice also exhibit deficits in EC-DG synaptic plasticity, impaired hippocampal learning, and spreading of tau into the DG [9,53–56], detailed examination has revealed ectopic tau expression outside of the EC [57]. Interestingly, even though tau is largely expressed in the MEC in these mice, it is the LEC that ultimately displays the greatest pathology [9].

To model the pattern of pathology observed in humans, with pathway- and cell-specific pathology, here we used a viral approach to localize human tau pathology specifically to the LEC. Using transgenic mice to birthdate DG neurons we then examined how tau pathology impacts morphometric and electrophysiological properties of new and old neurons in the LEC-DG circuit.

## Methods

### Animals and treatments

All procedures were approved by the Animal Care Committee at the University of British Columbia and conducted in accordance with the Canadian Council on Animal Care guidelines. Mice had ad libitum access to food and water and were housed on a 12 h light/dark cycle with lights on at 7 AM. The general experimental design and timeline is presented in Fig 1. Ascl1CreERT2 mice [58] were crossed with CAGfloxStopTdTomato mice [59] (Ai14; The Jackson Laboratory) to generate 82 male and female experimental mice whose precursor cells/newborn neurons could be induced to express tdTomato via tamoxifen injection [46,60–64]. All mice received recombinant adeno-associated virus (rAAV) injections at 3–6 months of age to express Tau-GFP or GFP in the LEC and lateral perforant pathway to the DG. Tamoxifen (Sigma; T5648) was injected either on postnatal day 1 (1 x 75 mg/kg, in sunflower oil) to label developmentally-born neurons (DBNs) or 3 months after rAAV injection (1 x 150 mg/kg), to label adult-born neurons (ABNs). Brains were harvested for morphological and electrophysiological analyses at 4 months after rAAV injection. Sample sizes are described in the results section for each experiment.

### rAAV production

Two vectors were used in this project: rAAV9-hTau40-Ubi-C-IRES-GFP and control virus rAAV9-Ubi-C-IRES-GFP. The human full length Tau is driven by the human Ubiquitin ligase C promoter followed by an internal ribosome entry site (IRES) driven expression of green fluorescent protein (GFP). rAAV were generated as previously described [65]. Briefly, HEK293 cells were co-transfected with either the pTR-GFP plasmid or the pTR-hTau-GFP plasmid, in addition to the pXX6 helper plasmid and pAAV9 serotype plasmid. Transfection was performed using polyethyleneimine reagent. The virus was harvested from cells and purified using an iodixanol gradient and concentrated using centrifugal filtration. Viral titer was determined with the dot blot assay described previously [65]. Vectors are described as vector genomes (vg)/mL.

### Stereotaxic surgery and injection of tau-expressing rAAV

Tau pathology was induced by locally expressing full length wild type human Tau (hTau; Tau40, 2N4R) in the LEC. Wild type, 4-repeat (4R) tau was chosen because sporadic age-related tau pathology is not associated with mutations in the

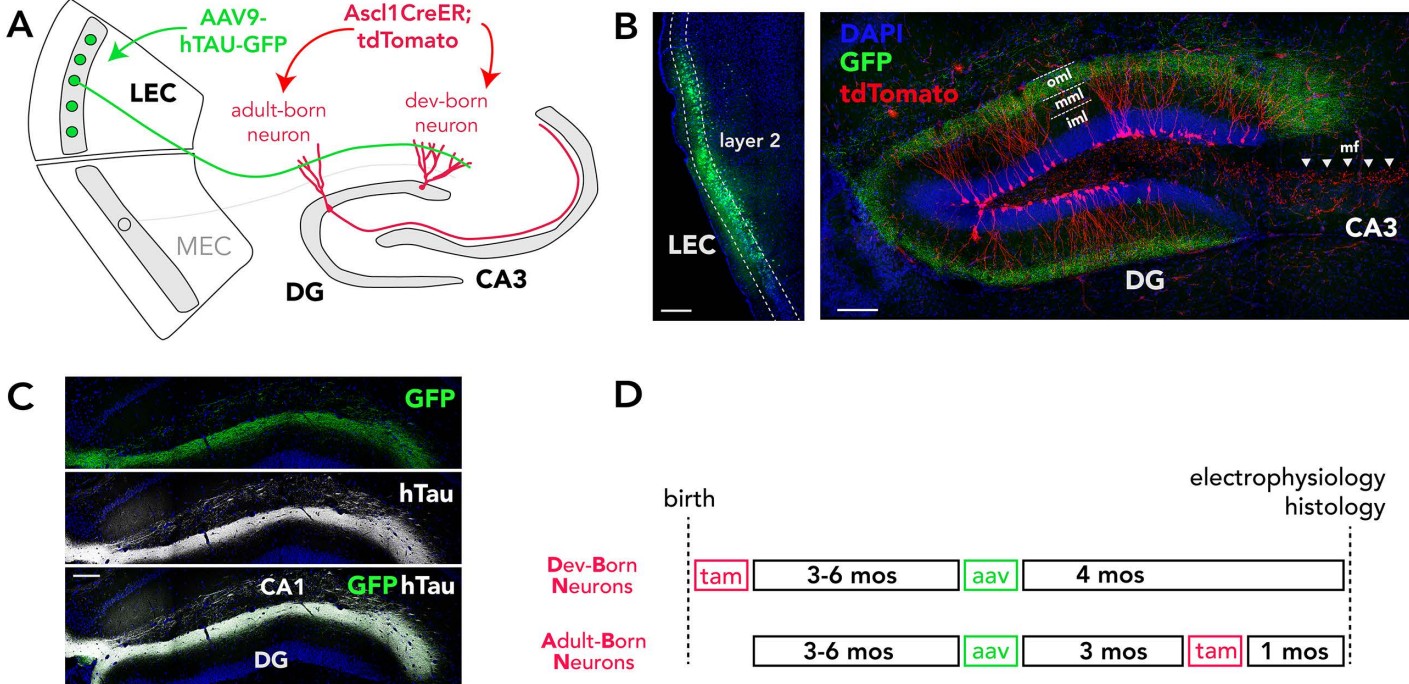

**Fig 1. Tau model and experimental design. A)** rAAV was injected into the LEC to express hTau-GFP or GFP in DG-projecting layer 2 neurons. tdTomato⁺ developmentally- or adult-born DG neurons were visualized with Ascl1CreER mice. **B)** Confocal images of GFP expression in layer 2 LEC neurons (left) and their corresponding axons (right), which target the outer molecular layer of the DG (tdTomato⁺ adult-born neurons shown). **C)** Overlapping GFP and hTau expression in the axons of LEC neurons that were transduced with rAAV-hTau-GFP. **D)** Experimental timeline: All mice were injected with rAAV at 3-6 months of age and were tested for electrophysiology and morphometric analyses 4 months later. Developmentally-born neurons were targeted by injecting tamoxifen on postnatal day 1 and adult-born neurons were targeted by injected tamoxifen 1 month before collecting brains. Scale bars 250µm. oml/mml/iml: outer, middle/inner molecular layer; mf: mossy fiber axons, tam: tamoxifen.

tau gene, and overexpression of 4R tau is observed in Alzheimer's disease [66]. rAAV9-hTau40-Ubi-C-IRES-GFP (rAAV-hTau-GFP; 9.6 x 10¹² vg/ml) or control virus rAAV9-Ubi-C-IRES-GFP (rAAV-GFP; 1.4 x 10¹³ vg/ml) was injected into the LEC of 6- to 9-month-old mice according to sterile surgical procedures (ages balanced across DBN and ABN groups). Mice were anaesthetized with isoflurane, given meloxicam (5 mg/kg, SQ), local bupivacaine (8 mg/kg, SQ) and lactated ringers solution (10 ml/kg) at the start of surgery. Following standard stereotaxic techniques, bilateral rAAV injections into the LEC were made at − 3.6 mm posterior, ± 4.7 mm mediolateral and −2.6 mm ventral relative to bregma. One µl of virus was injected into each hemisphere, at a speed of 200 nl/min, using a 30-gauge Hamilton needle and 10 µl Hamilton syringe with a microsyringe pump (World Precision Instruments). The needle remained in place for 5 min after the injection to allow for diffusion.

### Tissue processing and immunohistochemistry

Mice were transcardially perfused with 4% paraformaldehyde in phosphate buffered saline (PBS, pH 7.4) and brains remained in paraformaldehyde for 48 h prior to sectioning. Brain hemispheres were sectioned using a vibratome (Leica VT1000S), with one hemisphere cut at 100 µm for morphological analysis and the other cut at 40 µm for all other histological analyses. Brain sections were stored in cryoprotectant at −20°C until immunohistochemical processing.

For free-floating fluorescent immunostaining of red fluorescent protein (RFP; tdTomato), sections were first washed with PBS (3 x 5 min). The sections were incubated with a blocking solution of 3% horse-serum in 0.5% PBS-Triton X for 30 min,

and then incubated with rabbit anti-RFP antibody (1:2000 in blocking solution; Rockland 600-401-379) for 48 hours on a shaker plate at 4°C. Then tissue was washed with 0.5% PBS-Triton X and incubated with secondary antibody (1:250 donkey anti-rabbit Alexa 555) for 1 hr and then washed with PBS.

For free-floating fluorescent immunostaining of tau protein, sections were retrieved from the cryoprotectant solution and washed with PBS. Brain sections were incubated in 0.01M citric acid at 98 degrees C for 10 minutes. The tissue was then immersed in 3% horse-serum in 0.5% PBS-Triton X for 30 min, and incubated with either a rabbit polyclonal anti-human tau (hTau) antibody (Invitrogen PA527287), a phospho-tau monoclonal antibody to detect AT8 (Invitrogen MN1020), a rabbit polyclonal phospho-tau Ser396 (Invitrogen 44-752G) or chicken anti-GFP (Aves Labs GFP-1010) for 3 days on a shaker plate at 4°C. After the primary antibody incubation, slices were washed with 0.5% PBS-Triton X and incubated with their respective secondary antibodies for 1 h (1:250 donkey anti-mouse Alexa 647 antibody; 1:250 donkey anti-rabbit Alexa 647 antibody; 1:250 donkey anti-chicken Alexa 488 antibody) followed by a PBS wash for 5 min.

All sections were stained with DAPI in PBS (1:1000) to visualise cell nuclei. Brain slices were mounted on slides and cover-slipped with mounting medium PVA-DABCO to prevent fluorescent fading.

### Imaging and morphological analyses

Images were acquired with a Leica SP8 confocal microscope for all analyses. For confirmation of surgical hits and viral spread from the LEC to the DG, tau and GFP were imaged along the anterior-posterior axis using a 10X objective. Images were qualitatively assessed for anatomical spread and presence of hTau and phosphorylated Tau variants. For cell density quantification, images of the LEC region were collected at approximately −3.8 mm (AP) from Bregma with 40X oil immersion objective at 1024 x 1024 resolution along a 1 µm thick z-section in the middle of the stack at 1X zoom.

For morphological quantification, 100 µm thick sections labelled for RFP were imaged from the dorsal DG (−1.2 mm to −2.2 mm AP relative to Bregma). Dendritic branching and spine analyses were performed on cells located in the suprapyramidal blade from animals that had hTau in the LEC and outer molecular layer of the DG. To measure dendritic branching, images of RFP-labelled neurons in the DG (n = 3 per animal) were collected at 1024 x 1024 resolution with a z-stack step size of 1 µm with a 25X water objective (N.A. = 0.95) at 1X zoom. Dendrites were traced in ImageJ with the Simple Neurite Tracer plugin [67] to obtain dendritic length and number of branch points along dendritic tree throughout the 3D z-stack. Sholl analyses were performed to analyze complexity of dendritic branching [43,68].

For measuring dendritic spines, images were acquired with a glycerol-immersion 63X objective (N.A. = 1.40), at 1024 x 1024 resolution, 0.33 µm z-step size and 5X zoom. Dendritic segments from the same cells were sampled from the inner, middle and outer molecular layers to analyze whether spine counts varied in dendritic regions that receive hilar, medial entorhinal, and lateral entorhinal inputs, respectively. Thin spines (postsynaptic spines with thin neck and bulbous head) and mushroom spines (mature spines with a large head ≥ 0.6 µm in diameter) were counted with the ImageJ Cell Counter plugin and categorized according to established criteria [69]. All spine counts were normalized to the dendrite length.

For measuring large mossy fiber bouton (MFB) terminals of granule neuron axons, images were acquired with a glycerol-immersion 63X objective, at 1024 x 1024 resolution, 1 µm z-step size and 5X zoom. The MFBs were sampled along CA3 subregions (CA3a, CA3b, CA3c) with an average of 5 boutons measured per region, for a total of 15 boutons per animal. The area of each bouton was measured the maximum projection of the z-stack sections using ImageJ. MFB-associated filopodia protrusions, which contact inhibitory interneurons, were also counted and their lengths were measured.

### Cell counting

Quantification of DAPI-stained nuclei was performed in LEC layer II to determine cell loss under tau pathology. DAPI+ cells were counted between −3.6 to −3.8 mm from bregma (virus injection site) in one section per animal from a slice taken mid z-stack with ImageJ Cell Counter plugin. Tissue area was obtained by tracing the 2D area of the LEC layer II and cell

density was quantified by dividing cell counts by total area (to get cells per $mm^2$). For consistency, the ROI measured for cell quantification was similar in size across animals (average area = 0.16 $mm^2$).

## Brain slice preparation

Mice were anesthetized with sodium pentobarbital (50 mg/kg, I.P.) and were perfused with ice-cold cutting solution containing (in mM): 93 NMDG, 2.5 KCl, 1.2 $NaH_2PO_4$, 30 $NaHCO_3$, 20 HEPES, 25 glucose, 5 sodium ascorbate, 3 sodium pyruvate, 10 n-acetyl cysteine, 0.5 $CaCl_2$, 10 $MgCl_2$ (pH-adjusted to 7.4 with HCl and equilibrated with 95% $O_2$ and 5% $CO_2$, ~310 mOsm). Transverse hippocampal slices were cut on a vibratome and transferred to NMDG-containing cutting solution at 35ºC for 20 minutes, before being transferred to a storage solution containing (in mM): 87 NaCl, 25 $NaHCO_3$, 2.5 KCl, 1.25 $NaH_2PO_4$, 10 glucose, 75 sucrose, 0.5 $CaCl_2$, 7 $MgCl_2$ (equilibrated with 95% $O_2$ and 5% $CO_2$, ~325 mOsm) at 35ºC before starting experiments.

## Electrophysiology

Whole-cell patch-clamp recordings were made at near-physiological temperature (~32ºC) from identified tdTomato+ granule cells in the suprapyramidal blade of the dentate gyrus. Slices were superfused with an artificial cerebrospinal fluid (ACSF) containing (in mM): 125 NaCl, 25 $NaHCO_3$, 2.5 KCl, 1.25 $NaH_2PO_4$, 25 glucose, 1.2 $CaCl_2$, 1 $MgCl_2$ (equilibrated with 95% $O_2$ and 5% $CO_2$, ~320 mOsm). In all experiments GABAergic inhibition was blocked with bicuculline methiodide (10 uM). Recording pipettes were fabricated from 2.0 mm/ 1.16 mM (OD/ID) borosilicate glass capillaries and had resistance ~5 MOhm with an internal solution containing (in mM): 120 K-gluconate, 15 KCl, 2 MgATP, 10 HEPES, 0.1 EGTA, 0.3 $Na_2GTP$, 7 $Na_2$-phosphocreatine (pH 7.28 with KOH, ~300 mOsm). Current-clamp and voltage-clamp recordings were performed at −80 mV. Only recordings with high seal resistance (several giga-ohms) and low holding current (less than 50 pA) were included in analyses. For current-clamp recordings, series resistance and pipette capacitance were compensated with the bridge balance and capacitance neutralization circuits of the amplifier. A bipolar electrode was placed in the outer 1/3 of the molecular layer to stimulate the lateral perforant path (LPP) fibers. Stimuli (0.1 ms) were delivered through a stimulus isolator (A-M Systems analog stimulus isolator model 2200). Up to 3 experiments were performed on a single cell. First, synaptic transmission input-output curves were obtained by measuring EPSCs in response to afferent stimulation from 100µA to 1000µA. Then, paired-pulse facilitation was assessed using 50-Hz pairs of pulses in voltage clamp. Finally, LTP was measured in current clamp. For LTP experiments, single EPSPs (~3–5 mV) were evoked every thirty seconds before and after a single theta-burst stimulation (TBS) consisting of 10 trains of 10 pulses (100-Hz), delivered at 5-Hz, and repeated four times at 0.1 Hz, paired with postsynaptic current injection (100 pA, 100 ms). LTP was calculated as the increase in synaptic strength at 30–40 minutes post-TBS, with the exception of one cell that was calculated based on the increase from 25–40 minutes post-TBS due to occasional spiking that obscured EPSP size and necessitated a larger temporal window to obtain sufficient sampling. Finally, some cells did not last an entire recording session, resulting in fewer cells for the LTP experiment compared to the input-output curves and paired pulse analyses.

## Statistical analyses

All underlying data are provided as Supporting Information (S1 File). Analyses were performed in Prism 9.0 (Graphpad) and R. Cell counts were analyzed by unpaired t-test (DAPI) or sex x treatment ANOVAs (DCX). Morphological and electrophysiological properties of tdTomato+ cells were analyzed by fitting mixed effects models (using the lme4 package in R) to account for correlations between cells that were sampled from the same animal [70]. Sex and treatment (GFP vs hTau) were treated as fixed effects and subject was treated as a random effect. Main effects and interactions were then analyzed by ANOVA, followed by Sidak-adjusted post hoc comparisons and, in the case of input-output curves, simple slopes analyses with emtrends. Sholl data were similarly analyzed, as described [71] and adapted for R [72], with the exception

that both subject and cell were treated as random effects in the mixed models (cell nested within subject). All tests had significance set as alpha < 0.05.

## Results

### Tau expression and histopathology in the entorhinal-hippocampal axis

rAAV was injected into the LEC to induce hTau-GFP or GFP expression primarily in DG-projecting layer 2 neurons (Fig 1, 2A-C). A single injection resulted in widespread tau expression along the rostrocaudal and dorsoventral axes of the DG molecular layer, reflecting expression in perforant path axons. Only the very ventral DG failed to show expression (Fig 2A). Expression was always strongest in the lateral perforant path, which targets the distal dendrites of DG granule neurons. In some mice there was also limited expression in the temperoammonic pathway targeting distal CA1 dendrites, which originates in LEC layer 3 neurons (Fig 2B). As expected, no hTau expression was observed in mice that were injected with rAAV-GFP (Fig 2C). The pathological effects of tau are, in part, the result of increased phosphorylation and aggregation into paired helical filaments [73]. We therefore examined whether our model led to expression of these pathological forms of tau. Indeed, immunostaining with a phospho-S396 targeting antibody confirmed the presence of phosphorylated tau in the LEC and in the perforant pathway projection to the DG (Fig 2D). Similarly, AT8 immunoreactivity revealed paired helical filament tau species in both the LEC and perforant path projection (Fig 2E). Finally, we quantified cell density in LEC layer 2 since neurons in this layer are particularly vulnerable in human aging and the earliest stages of Alzheimer's disease. We found no significant difference between mice injected with rAAV-hTau-GFP vs rAAV-GFP ($T_{19} = 0.45$, $P = 0.66$). This suggests that our model may be most reflective of the earliest stages of age-related cognitive decline, when pathological tau has begun to be expressed in EC circuits but before any cell death has occurred.

### Neurogenesis in hTau-expressing circuits

Adult neurogenesis is reduced in most amyloid-based mouse models of Alzheimer's disease [74] and human neurogenesis levels, as measured by DCX expression, inversely correlates with the severity of Alzheimer's disease [75]. To determine whether neurogenesis is altered in our model of early entorhinal tau pathology, we quantified immature DCX$^+$ cells in mice injected with rAAV-GFP (N = 10 male, 5 female) vs rAAV-hTau-GFP (N = 6 male, 6 female). Here we found that presynaptic hTau expression was associated with ~30% more DCX$^+$ neurons (Fig 3). Neurogenesis was also higher in male mice than in female mice (effect of sex: $F_{1,20} = 4.9$, $P = 0.039$; effect of hTau: $F_{1,20} = 6.2$, $P = 0.022$; interaction: $F_{1,20} = 0.6$, $P = 0.44$).

### LEC hTau reduces dendritic complexity in ABNs from male mice

The morphological development of ABNs is highly sensitive to experience, activity and pathology. To test whether functionally-relevant morphological properties of ABNs are affected by presynaptic tau pathology we first analyzed the dendritic structure of developing, ~1-month-old ABNs (GFP mice: N = 22 male, 14 female; hTau mice: N = 29 male, 15 female; labelling was too dense to permit accurate dendritic reconstruction of DBNs). We first examined the total number of dendritic branches and found that this was not altered by hTau (effect of hTau: $F_{1,25} = 1.1$, $P = 0.3$, effect of sex: $F_{1,25} = 0.1$, $P = 0.7$; interaction: $F_{1,25} = 0.9$, $P = 0.4$). Since Alzheimer's pathology has been found to alter the primary dendrites of ABNs [76] we also measured primary dendrite length but found no differences (effect of hTau: $F_{1,25} = 0.1$, $P = 0.8$, effect of sex: $F_{1,25} = 0.9$, $P = 0.4$; interaction: $F_{1,25} = 0.1$, $P = 0.7$). Next, we measured total dendritic length and found no effect of hTau ($F_{1,27} = 0.6$, $P = 0.5$) or sex ($F_{1,27} = 0.1$, $P = 0.8$; Fig 4B). However, while the hTau x sex interaction was not statistically significant ($F_{1,27} = 3.8$, $P = 0.06$), exploratory analyses suggest that hTau may differentially impact dendrites in males (17% reduction in length, $P = 0.04$) vs females (7% increase, $P = 0.4$).

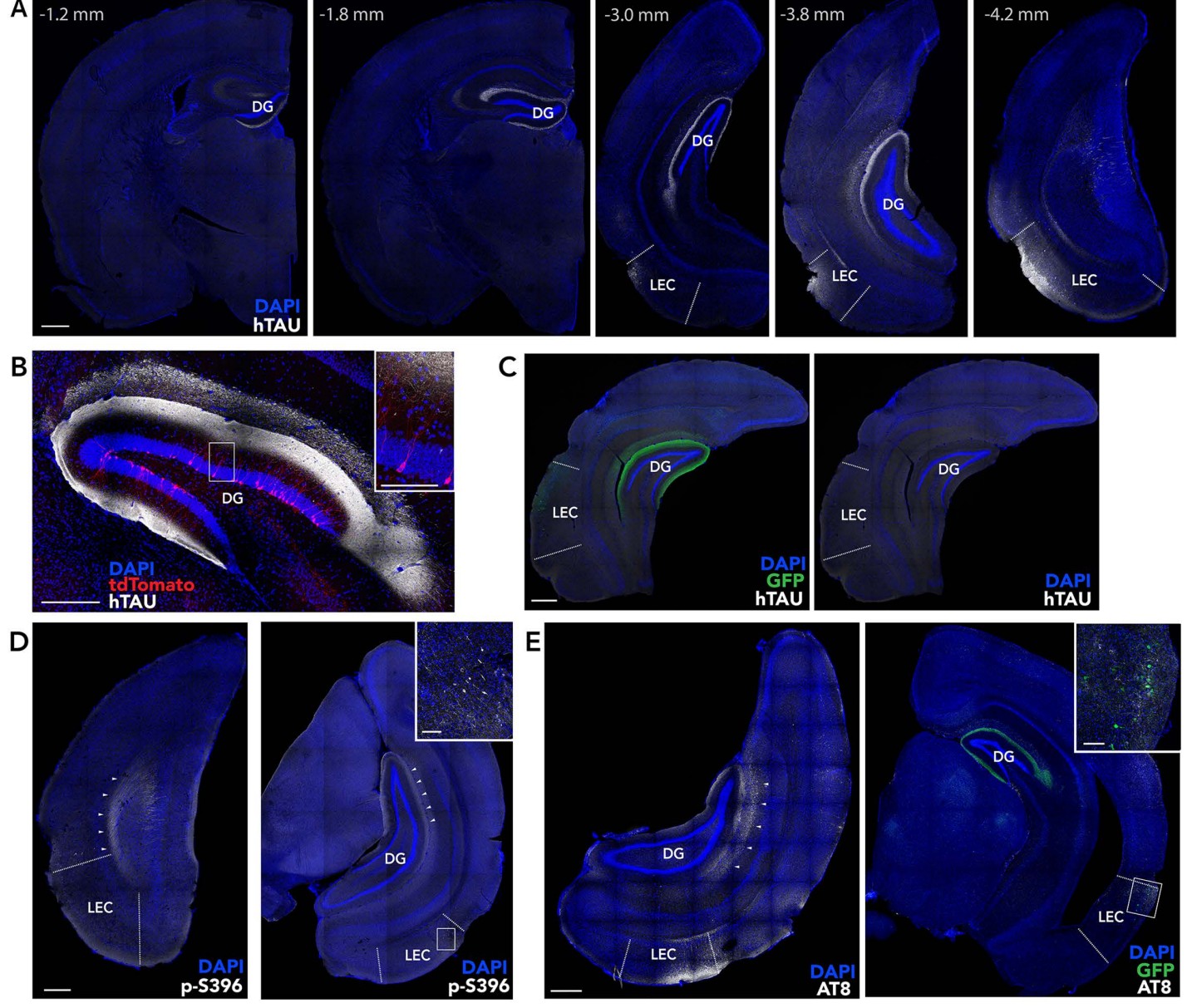

**Fig 2. Tau pathology in the LEC-DG pathway. A)** Anterior-posterior expression of human tau in the LEC and lateral perforant pathway projection to the outer molecular layer of the DG. **B)** Adult-born tdTomato⁺ DG granule cells in a mouse injected with rAAV-hTau-GFP. **C)** Absence of human tau in a control mouse injected with rAAV-GFP. **D)** Phosphorylated tau immunostaining, with p-S396 antibody, of LEC neurons and perforant path axons (arrowheads) in a mouse injected with rAAV-hTau-GFP. **E)** Phosphorylated paired helical tau, immunostained with AT8 antibody, in the LEC and perforant pathway (arrowheads). Scale bars: 500 μm (low mag), 100 μm (high mag insets).

We hypothesized that presynaptic hTau at distal dendrites could alter the distribution of dendritic branching. We therefore performed a Sholl analysis to quantify dendritic branching at various distances from the soma. Again, we fit the data with mixed-effects models to account for correlations in the data that result from analyzing multiple cells from the same animal, and analyzing the same cells at multiple distances from the soma [71,72]. An initial analysis, with hTau, sex and distance as fixed effects, revealed a significant effect of hTau ($F_{1,2472} = 14$, $P = 0.0002$), no effect of sex ($F_{1,2472} = 2.2$,

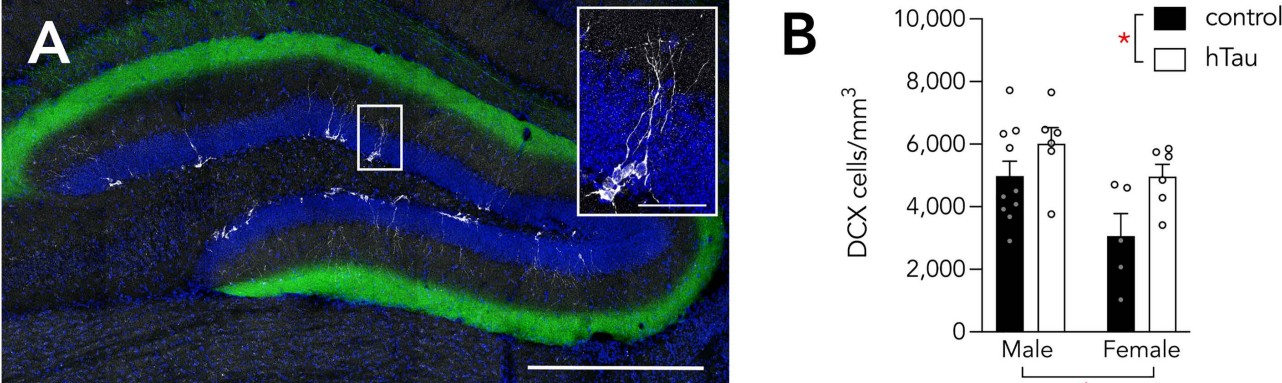

**Fig 3. Neurogenesis is elevated hTau-expressing mice. A)** DCX+ immature neurons in a mouse injected with GFP control rAAV. Scale bar 500 μm, 50 μm (inset). **B)** DCX+ cell density was greater in males than in females and greater in the hTau group than in GFP controls. Bars indicate mean ± standard error. *P < 0.05.

P = 0.13) and a significant hTau x sex interaction ($F_{1,2472} = 36$, $P < 0.0001$) suggesting that hTau has different effects on dendritic complexity in males and females. We therefore separately analyzed dendritic complexity in males and females and found that, in females, there was no effect of hTau on dendritic complexity ($F_{1,943} = 0.6$, $P = 0.43$; Fig 4C). In contrast, in males, hTau significantly reduced the number of dendritic intersections ($F_{1,1394} = 51$, $P < 0.0001$), particularly in dendritic branches located in the molecular layer ~160–260 μm from the soma. While sample sizes were not equivalent across the sexes, it is unlikely that the absence of a hTau effect in females is entirely due to a lack of statistical power since there was no apparent trend for a group effect.

### hTau induces spine loss and growth in ABNs

To determine whether presynaptic hTau specifically affects post-synaptic sites we quantified the density of thin spines (which make up the majority of dendritic protrusions) on ABNs and DBNs, from GFP and hTau mice (ABNs: average 37 cells/mouse, 7 mice per group; DBNs: 11 cells/mouse, 3 mice per group). In ABNs from both male and female mice, hTau caused an increase in the overall number of thin spines (Fig 5B; effect of hTau: $F_{1,28} = 14$, $P = 0.001$; effect of sex: $F_{1,28} = 1.7$, $P = 0.2$, interaction: $F_{1,28} = 0.01$, $P = 0.92$). This effect was similar in the inner (primarily associational/commissural inputs), middle (primarily MEC inputs), and outer (primarily LEC inputs) molecular layers (sexes pooled; layer x hTau interaction: $F_{2,278} = 0.5$, $P = 0.6$). We next examined DBNs to determine to explore the generality of this effect. Here, we failed to find any effect of hTau on thin spine density (Fig 5C; effect of hTau: $F_{1,10} = 3.7$, $P = 0.8$; effect of sex: $F_{1,10} = 0.1$, $P = 0.7$, interaction: $F_{1,10} = 2.3$, $P = 0.2$). There was also no effect of hTau on DBNs when examined as a function of molecular layer subregion (sexes pooled; layer x hTau interaction: $F_{2,92} = 1$, $P = 0.4$). Unfortunately, we were unable to sample as many DBNs, due to the high density of tdTomato labelling. Nonetheless, these data suggest that ABNs may have more compensatory plasticity than DBNs in response to presynaptic hTau.

Thick, mushroom-shaped spines are more stable over time and may reflect long-term memory substrates [69,77]. We therefore analyzed the density of mushroom spines in ABNs and DBNs. In contrast to the findings for thin spines, above, here we found a consistent decrease in the density of mushroom spines in hTau mice regardless of when the cell was born (Fig 5D,E; ABNs: effect of hTau: $F_{1,28} = 13$, $P = 0.001$; effect of sex: $F_{1,28} = 0.2$, $P = 0.7$, interaction: $F_{1,28} = 0.4$, $P = 0.5$; DBNs: effect of hTau: $F_{1,10} = 12$, $P = 0.01$; effect of sex: $F_{1,10} = 0.1$, $P = 0.7$, interaction: $F_{1,10} = 0.8$, $P = 0.4$). Notably, even though hTau was mainly located in the lateral perforant path, the loss of mushroom spines was equivalent across

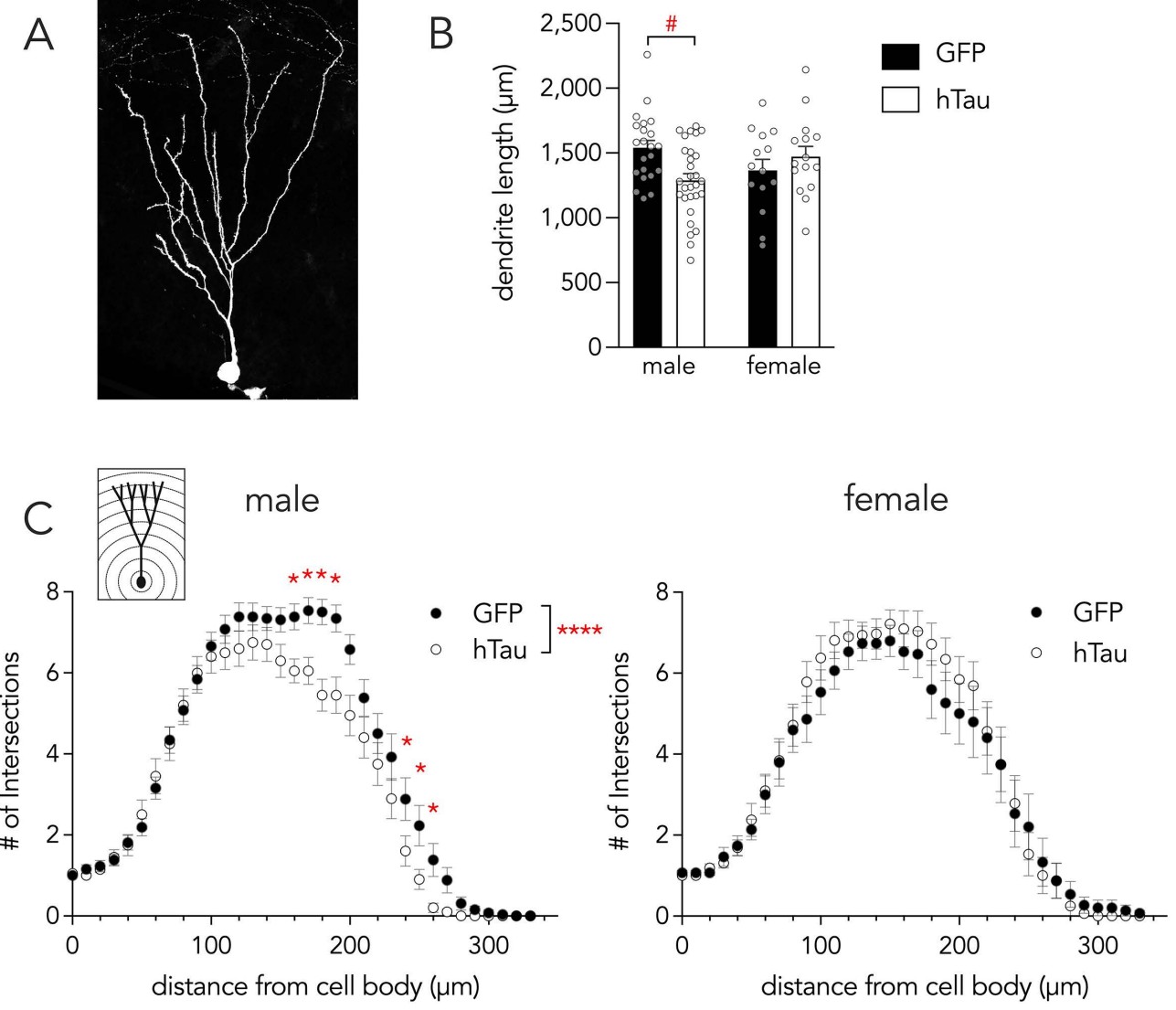

**Fig 4. hTau reduces dendritic complexity of adult-born neurons in male mice. A)** Maximum projection image of a tdTomato+ ABN for dendritic analyses. **B)** Exploratory analyses suggest hTau may reduce dendritic in males. **C)** Sholl analysis revealed reduced dendritic complexity in male mice expressing hTau, but not female mice. Bars and symbols indicate mean±standard error. *P<0.05, ****P<0.0001, #P<0.05 (exploratory analysis; hTau x sex interaction P=0.06).

molecular layer subregions in both ABNs and DBNs (sexes pooled; ABNs: layer x hTau interaction: $F_{2,278}=0.9$, P=0.4; DBNs: layer x hTau interaction: $F_{2,60}=0.8$, P=0.4).

## LEC hTau reduced mossy fiber terminal size in CA3c

We did not observe tau spreading beyond the EC-DG circuit. Nonetheless, to determine whether hTau-induced morphological changes are observed at sites that are downstream from the entorhinal input synapses we examined the structural properties of the output mossy fiber terminals of ABNs and DBNs (ABNs: average 32 boutons from each of 6–9 mice/sex/treatment; DBNs: average 21 boutons from each of 3–7 mice/sex/treatment). We first examined presynaptic terminal size,

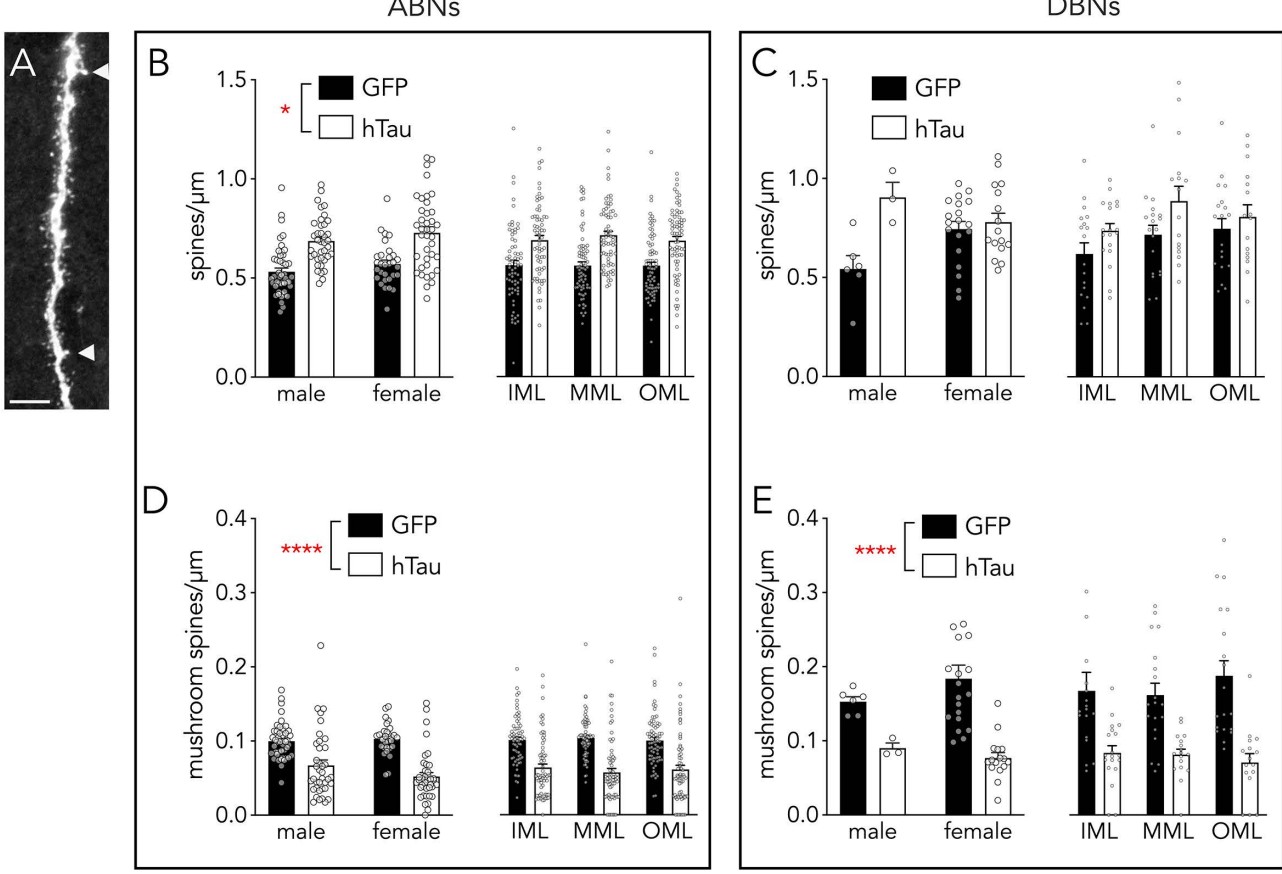

**Fig 5. Effects of tau overexpression on dendritic spines for ABNs and DBNs. A)** Confocal image of a dendritic segment from an ABN. Mushroom spines indicated by arrowheads; scale bar = 5 μm. **B)** Presynaptic hTau increased thin spine density in ABNs in both male and female mice, and in all regions of the molecular layer. **C)** hTau did not alter thin spine density in DBNs. **D)** hTau decreased mushroom spine density in ABNs from both male and female mice, and in all regions of the molecular layer. **E)** hTau decreased mushroom spine density in DBNs from both male and female mice, and in all regions of the molecular layer. IML, MML, OML refers to inner, middle and outer molecular layer. Bars indicate mean ± standard error. *P < 0.05, ****P < 0.0001.

which serves as a proxy for synaptic strength [78]. In ABNs, we found no significant differences between boutons from mice treated with the hTau-expressing rAAV vs GFP controls (Fig 6B; ABNs: effect of hTau: $F_{1,25} = 0.8$, P = 0.4, effect of sex: $F_{1,25} = 0.1$, P = 0.7, sex x hTau interaction: $F_{1,25} = 0.4$, P = 0.5). In DBNs we also observed no significant effects of hTau on bouton size (Fig 6C; effect of hTau: $F_{1,12} = 0.2$, P = 0.7, sex x hTau interaction: $F_{1,12} = 2$, P = 0.2). However, we did find that boutons were larger in females than in males, consistent with previous findings [79] (effect of sex: $F_{1,12} = 8$, P = 0.02). Since CA3a/b/c subregions have distinct functions in pattern separation and completion [80], and since there were no hTau-related sex differences, we pooled male and female data for treatment x CA3 subregion analyses. Here we found that in ABNs there was a significant hTau-related reduction in bouton size in CA3c, whereas there were no changes in DBNs (ABNs: hTau x subregion interaction: $F_{2,380} = 6.2$, P = 0.002; CA3a: GFP vs hTau $T_{62} = 0.02$, P = 0.98; CA3b: $T_{58} = 0.8$, P = 0.45; CA3c: $T_{54} = 2.6$, P = 0.01; DBNs: hTau x subregion interaction: $F_{2,258} = 0.4$, P = 0.7). This suggests that ABN terminal changes may impact CA3c functions in pattern separation [80].

Mossy fiber bouton filopodial extensions are presynaptic terminals that innervate inhibitory interneurons [81] and contribute to the sparse hippocampal coding necessary for memory [82,83]. We therefore quantified the length

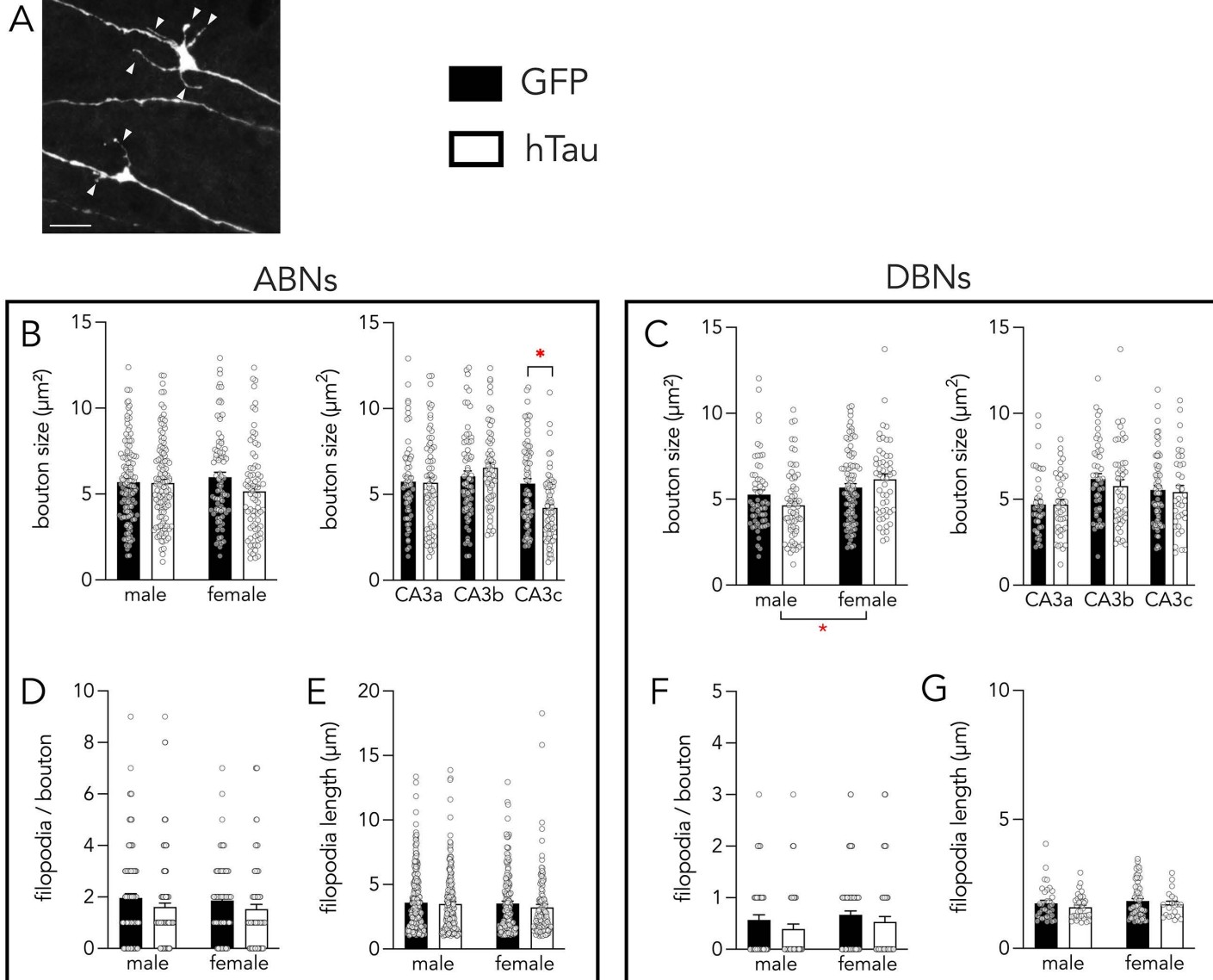

**Fig 6. Effects of hTau on mossy fiber boutons. A)** Confocal image of a large bouton with 5 filopodia and a small bouton with 2 filopodia (indicated by arrowheads). Scale bar = 5μm. **B)** hTau did not induce overall changes in ABN bouton size in male or female mice (left) but a regional analysis did reveal smaller boutons in CA3c (right; sexes pooled). **C)** hTau did not alter bouton size in DBNs. **D)** hTau did not alter the number of filopodia per bouton in ABNs. **E)** hTau did not alter the length of filopodia in ABNs. **F)** hTau did not alter the number of filopodia per bouton in DBNs. **G)** hTau did not alter the length of filopodia in DBNs. Bars indicate mean ± standard error. *P < 0.05.

and frequency of bouton-associated filopodia on ABNs and DBNs (Fig 6D-G). For both measures, we found no hTau-related differences (Filopodia length: ABNs: effect of hTau: $F_{1,26} = 1$, $P = 0.3$, effect of sex: $F_{1,26} = 0.6$, $P = 0.5$, interaction: $F_{1,26} = 0.1$, $P = 0.8$; DBNs: effect of hTau: $F_{1,15} = 0.4$, $P = 0.5$, effect of sex: $F_{1,15} = 0.4$, $P = 0.6$, interaction: $F_{1,15} = 0.3$, $P = 0.6$. Filopodia per bouton: ABNs: effect of hTau: $F_{1,25} = 1.4$, $P = 0.2$, effect of sex: $F_{1,25} = 0.1$, $P = 0.8$, interaction: $F_{1,25} = 0.01$, $P = 0.9$; DBNs: effect of hTau: $F_{1,17} = 0.6$, $P = 0.5$, effect of sex: $F_{1,17} = 0.001$, $P = 1$, interaction: $F_{1,17} = 0.5$, $P = 0.5$).

## Effects of LEC hTau on synaptic transmission and plasticity at lateral perforant path input synapses

The morphological analyses pointed to differences primarily in afferent structures (i.e., spines and dendrites), therefore we examined the strength of hTau-expressing LEC input synapses in the outer molecular layer (ABNs: 9 cells from 7 GFP mice, 16 cells from 11 hTau mice; DBNs: 13 cells from 8 GFP mice, 13 cells from 9 hTau mice).

In ABNs, there was an expected increase in EPSC magnitude as stimulation intensity increased (Fig 7A). While there was no overall effect of hTau, significant 2-way and 3-way interactions suggested hTau effects differed as a function of stimulus intensity and sex (effect of stimulation intensity: $F_{1,221} = 139$, P < 0.0001; effect of hTau: $F_{1,16} = 0.001$, P = 1; effect of sex: $F_{1,17} = 0.1$, P = 0.9; hTau x stimulus intensity interaction: $F_{1,221} = 7$, P = 0.007; hTau x sex interaction: $F_{1,17} = 0.02$, P = 1; hTau x sex x stimulation intensity interaction: $F_{1,221} = 13$, P = 0.0004). Pairwise comparisons between GFP and hTau mice (sexes pooled) at each stimulation intensity did not reveal any differences (all Ps > 0.16) but a simple slopes analysis showed that, with greater stimulation intensity, EPSCs increased in size to a greater extent in hTau mice as compared to GFP mice ($T_{1,221} = 3$, P = 0.007). To probe the hTau x sex x stimulus intensity interaction we first conducted within-sex pairwise comparisons between GFP and hTau mice at each stimulation intensity. This revealed no significant differences in either males (all P ≥ 0.05) or females (all P > 0.6), though responses at 800–1000 μA approached significance in males (0.05 < P < 0.10). However, a simple slopes analyses revealed that male hTau mice had a greater synaptic response with increasing stimulation intensities ($T_{221} = 4.0$, P = 0.0001; Fig 7Ai). In contrast, the slope of the stimulus-response curve in female mice was similar between GFP and hTau treatments ($T_{221} = 0.7$, P = 0.5; Fig 7Aii).

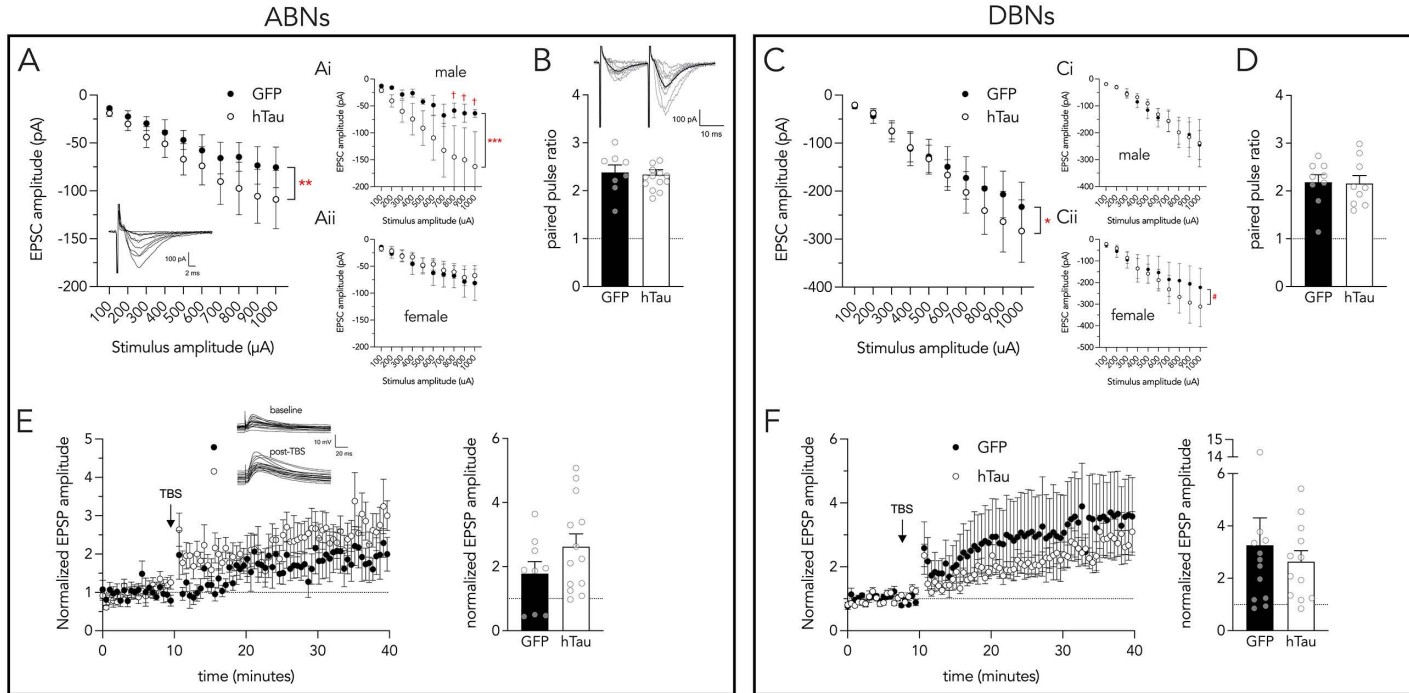

**Fig 7. LEC-DG synaptic transmission and plasticity. A)** Stimulus response curves were steeper in ABNs from hTau mice as compared to GFP mice (sexes pooled). Broken down by sex, ABNs from hTau males (Ai) but not females (Aii) have steeper stimulus intensity-EPSC slopes. **B)** Paired pulse ratios were not different between ABNs from GFP and hTau mice. **C)** Greater stimulation intensity-EPSC slope in DBNs from hTau mice. Broken down by sex, stimulus intensity-EPSC slopes were unaffected by hTau in males (Ci) but exploratory analyses suggest they were steeper in females (Cii). **D)** Paired pulse ratios were not different between DBNs from GFP and hTau mice. **E)** LTP was comparable in ABNs from mice expressing GFP and hTau. **F)** LTP was comparable in DBNs from mice expressing GFP and hTau. Bars and symbols indicate mean ± standard error. †0.05 < P < 0.10, *P < 0.05, **P < 0.01, ***P < 0.001, #P < 0.05 (exploratory analysis; 3-way interaction P = 0.06).

In DBNs there also was a clear effect of stimulation intensity ($F_{1,230}=263$, $P<0.0001$; Fig 7C) but no overall effects of hTau ($F_{1,13}=0.2$, $P=0.7$) or sex ($F_{1,13}=0.2$, $P=0.7$). There was no hTau x sex interaction ($F_{1,14}=0.5$, $P=0.5$) but there was a hTau x stimulation intensity interaction ($F_{1,230}=4$, $P=0.048$). Pairwise comparisons between GFP and hTau mice at each stimulation intensity did not reveal any differences (all Ps > 0.5). However, a simple slopes analysis showed that, with greater stimulation intensity, EPSCs increased in size to a greater extent in hTau mice as compared to GFP mice ($F_{1,230}=4$, $P=0.048$). While the hTau x sex x stimulation intensity interaction was not significant ($F_{1,230}=4$, $P=0.06$), exploratory simple slopes analyses suggest that the hTau x stimulation intensity interaction was primarily driven by females ($T_{230}=3$, $P=0.003$; Fig 7Cii) rather than males ($T_{230}=0$, $P=1$; Fig 7Ci). That is, as stimulation intensity increased, there may be a greater increase in EPSC size in female hTau mice than female GFP mice.

In our model, hTau is expressed presynaptically and previous work has found that the P301L mutation, when expressed in the medial entorhinal cortex, induces presynaptic deficits [84]. We therefore examined paired pulse ratios, which can be used to assess changes in transmitter release probability and short-term synaptic dynamics [85]. As expected for LEC synapses, which have low probability of initial release [86–88], paired pulse (50 Hz) responses were facilitating. They were also not different in either ABNs or DBNs, or as a function of sex or treatment (ABNs: effect of hTau: $F_{1,20}=0.0$, $P=0.8$; effect of sex: $F_{1,20}=0.1$, $P=0.8$; interaction: $F_{1,20}=1.6$, $P=0.2$; DBNs: effect of hTau: $F_{1,11}=0.8$, $P=0.4$; effect of sex: $F_{1,11}=0.1$, $P=0.7$; interaction: $F_{1,11}=1$, $P=0.3$). These data indicate no major changes in short-term presynaptic dynamics as a result of hTau expression.

To test whether hTau altered long-term synaptic plasticity, which could contribute to memory impairments in aging and Alzheimer's disease, we delivered high-frequency theta burst stimulation to the lateral perforant path and recorded EPSPs in current clamp. Both ABNs and DBNs displayed robust long-term potentiation, which did not differ between GFP- and hTau-expressing mice (Fig 7E,F; ABNs: effect of hTau: $F_{1,15}=1.1$, $P=0.3$, effect of sex: $F_{1,20}=0.2$, $P=0.7$, sex x hTau interaction: $F_{1,20}=0.0$, $P=0.8$; DBNs: effect of hTau: $F_{1,12}=0.9$, $P=0.4$, effect of sex: $F_{1,12}=2$, $P=0.2$, sex x hTau interaction: $F_{1,12}=1$, $P=0.3$).

## Discussion

Here we tested how a mouse model of early sporadic LEC Tau pathology affects structural and physiological properties at the LEC-DG pathway. By targeting human Tau40 (2N/4R) to the LEC we induced preferential expression in the lateral perforant pathway to the DG, which was associated with tau phosphorylation and AT8-immunoreactivity for paired helical tau. While our LEC cell density measurements did not differentiate between neurons vs glia, or between neuronal subtypes that specifically project to the DG, there clearly was no massive cell loss as is apparent in the LEC of individuals with AD and MCI [18,19]. Along with the lack of tau spreading into other brain regions, our hTau AAV approach would appear to most closely model very early stages of pathological aging. In terms of downstream effects, LEC hTau caused a mix of neurodegenerative and neuroplastic effects. With respect to neurodegenerative effects, hTau reduced dendritic complexity in male ABNs, reduced mushroom spines in ABNs and DBNs from both sexes, and caused general atrophy of ABN presynaptic terminals in CA3c. In terms of neuroplasticity, in both sexes hTau increased adult neurogenesis and numbers of thin spines, which comprise the majority of protrusions on DG granule cells. hTau also increased the synaptic input-output relationship in ABNs from male mice and in DBNs generally (possibly driven by females). Thus, in the face of modest presynaptic hTau pathology, downstream DG neurons display some signs of degeneration but also react with growth-associated plasticity and synaptic compensation, in part through adult neurogenesis.

### The role of LEC tau in pathological aging

Aging is associated with a stereotyped pattern of pathology and cognitive decline, where the lateral entorhinal cortex, and associated functions in memory, are among the first to deteriorate [9,32,34,89]. Even in mild cognitive impairment and the earliest stages of AD, the majority of entorhinal layer 2 neurons have died [18,19]. These cells provide the primary

input to the hippocampus and therefore their loss deprives the hippocampus of sensory information needed to encode and retrieve memories. However, it is likely that entorhinal pathology begins long before frank cell loss. In rodents, lateral perforant path LTP declines as early as 6–9 months of age (i.e., early middle age) [36]. In humans, histological staining for AT8 immunoreactivity has revealed a pattern whereby misfolded tau appears in the transentorhinal cortex as early as young adulthood [8]. While it is debated whether early abnormal tau is a fundamental component of Alzheimer's disease [90,91], the early rise in tau is ultimately toxic to neurons and synapses, in both animals [76,92] and humans [93–95], and is a prominent component of early pathological aging.

## Degeneration and plasticity in LEC hTau mice

Our observation of increased neurogenesis, morphological plasticity and stronger synapses (in DBNs, and in ABNs from males) would appear to conflict with previous reports that abnormal tau has primarily neurodegenerative effects including reducing neurogenesis, inducing cell death and atrophy of dendrites and synaptic structures, and causing deficits in synaptic strength and plasticity [53,76,92,96–99]. We propose that this discrepancy is because our model reflects a less aggressive and/or earlier stage of pathology, where expression is restricted to the LEC as compared to broader/brain-wide expression. Our model may more closely mimic the pattern of human early sporadic tau pathology, and our data points to potential for substantial compensatory plasticity. This is consistent with evidence that entorhinal damage in animals and Alzheimer's patients induces reactive innervation of the DG by entorhinal, commissural and sepal afferents [100,101]. In mouse models of amyloid pathology, there are also examples of compensatory growth of new spines, particularly when animals are younger [102,103]. While there is evidence that tau can lead to an increase in both the formation and loss of spines [98], here we provide new evidence that spine plasticity may lead to a net increase in spine numbers early in pathological aging, and an increase in synaptic strength (depending on sex and when the neuron was born). While our limited sampling of DBNs may have prevented us from detecting compensatory spinogenesis, the apparently selective plasticity of ABNs aligns with findings that ABNs in rodents ultimately have 60% more spines than DBNs[43]. An alternative hypothesis is that the elevated neurogenesis and spine numbers may be a persistent feature of the model as opposed to a compensatory form of plasticity that is limited to the early stages of pathology (as seen in an amyloid model [104]). It is also worth considering that tamoxifen injection itself may have affected plastic and neurogenic responses to hTau. While all animals received tamoxifen, effects on DG neurons may [105,106] or may not [107] differ between animals that received neonatal vs adult injections, and it is also possible that neuroprotective and anti-inflammatory effects of tamoxifen [108,109] could have mitigated the toxic effects of hTau on the EC-DG circuit [110]. This is most relevant for ABN mice since they received tamoxifen after hTau AAV injections. Future experiments are needed to test these possibilities.

Compensatory plasticity is attractive from a regenerative/therapeutic perspective, but it is important to consider the possibility that spine growth and increased synaptic strength might instead be detrimental to hippocampal function. For example, in aged rats [111] and humans [32] there is hyperactivity in the aged hippocampus, particularly in DG/CA3, which is associated with impaired performance on mnemonic discrimination tasks. Pharmacologically suppressing this hyperactivity in patients with mild cognitive impairment is sufficient to restore memory [112]. Thus, if ABN spinogenesis and enhanced LEC-DG synaptic input-output curves promote hippocampal activity, this may in fact impair learning and memory. We feel that this is unlikely, however, since hTau increased neurogenesis and new neurons often exert a net inhibitory effect on the DG/CA3 [113–117], protect against hyperactivity in other models of pathology [118], and so they are likely more suited to preserve hippocampal function in aging.

The neurodegenerative effects on mushroom spines and dendrites (in male ABNs) may ultimately relate to deficits in object-related learning and memory that are seen in human aging [32,34,35,119] and in rodent studies of the LEC, DG and neurogenesis [44,120–124]. Another mechanism by which tau may contribute to the early decline of detailed hippocampal memory is the selective atrophy of ABN mossy fiber terminals in CA3c. In this subregion, pyramidal neurons perform pattern separation rather than pattern completion seen elsewhere in CA3 [80]. Collectively, our data suggest a

thorough examination of behavior is needed to determine how the various forms of growth and atrophy of DG neurons contribute to the preservation and deterioration of hippocampal functions in memory.

## Sex differences in the response to hTau

It is well documented that AD is more prevalent in females than in males [6,7]. Fluid [125], PET [12–16] and post-mortem histology [126,127] indicate that women have a greater tau burden and/or greater association between tau and neuropathology or cognitive decline. Here, hTau had some effects that were comparable in males and females, namely increased neurogenesis, more thin spines, and fewer mushroom spines. When we observed sex differences it was new neurons in male mice that showed dendritic atrophy and steeper stimulus-response curves. Given that female human are more vulnerable, it is perhaps counterintuitive that male mice showed more hTau-related alterations. However, in the tau P301S model, it is also the male mice that display more pathological changes [128]. Also, in response to other perturbations, such as transcranial magnetic stimulation [79] and stressful learning [129], we have found greater morphological plasticity in ABNs from male rodents, suggesting possible sex differences in neural plasticity vs stability. Notably, amyloid was not a feature of our model and, in mouse models that incorporate both amyloid and tau, females tend to have greater overall pathology and cognitive impairment [130–132]. This pattern appears to hold for humans as well, where females are more vulnerable to the interactive effects of amyloid and tau [12,15,125,133]. Thus, experiments that examine the combinatorial effects of amyloid and tau are needed to fully disentangle the role that ABNs and DBNs play in age-related medial temporal lobe pathology.

## Relevance of neurogenic plasticity for humans

The birth of new DG neurons declines dramatically with age in all mammals and, while there are initial species differences in neurodevelopmental timing, neurogenesis in both humans and rodents plateaus to comparable levels for much of the lifespan [37]. Therefore the important question is whether there is sufficient neurogenic plasticity to offset pathological aging. To our knowledge, no one has directly tested whether neurogenesis contributes to behavior in aging; one study has manipulated neurogenesis in middle age but all others have manipulated neurogenesis in adolescence or young adulthood [37]. Nonetheless, it is estimated that there is 0.04% of daily cell addition in aging, which is sufficient to add ∼15% of DG neurons over the course of a decade, a number that may be relevant for combating the progression of age-related pathology [37]. New neurons compete with older neurons for synaptic space [134–137] and also appear to replace them as a part of physiological aging [138–140]. Thus, enhanced neurogenesis and spinogenesis could serve to replace synapses from DBNs that may be culled from hippocampal circuitry.

Another factor to consider is the possible long-term contribution of new neurons beyond the traditional (rodent) 4–6 week critical period of plasticity. Using zif268 as a marker of activity, adult-born neurons can be recruited during learning even when they are 19 months old, at rates that appear higher than embryonic-born neurons [141]. Adult-born neurons also develop morphologically over 6 months (∼25% of the lifespan) [43] and, unlike other (presumably developmentally-born) granule neurons, they undergo experience-dependent dendritogenesis and spinogenesis even when they are several months old [142,143]. This extended plasticity may be particularly relevant for offsetting age-related LEC pathology since there are multiple lines of evidence that adult-born neurons preferentially connect with the LEC [44,45]. Functionally, LTP at their LEC also develops over a much longer interval (months) than inputs from the medial entorhinal cortex [46]. Given evidence that new neurons mature much slower with age [144,145] and in longer-lived mammals such as primates [146], there may be a substantial reserve of neurogenic plasticity in humans, even after proliferation rates have declined. While evidence from humans is limited, it is notable that human DG neurons appear to continue to grow dendrites well into old age [147,148]. Immature DCX⁺ neurons are present in the aging brain but are reduced in Alzheimer's disease [75] and a recent transcriptomics study suggests that in humans there is a pool of immature neurons that persists into aging [149]. It remains unclear whether these examples of human DG plasticity reflect adult-born neurons, later-born neurons, or merely

old neurons that maintain a persistently immature phenotype. However, it points to a population of plastic neurons that may be comparable to those that we have identified here, which engage in a synaptogenic response to presynaptic Tau and may help preserve the functionality of this pathway in aging. A first step towards addressing this question might be to include behavioral analyses of LEC-DG function, and to examine aged mice, or mice with more severe tau pathology (e.g., longer rAAV-testing intervals).

## Supporting information

**S1 File. Full dataset.** This file contains all of the underlying data used for the graphs and analyses.
(XLSX)

## Author contributions

**Conceptualization:** Alyssa M. Ash, Kevin R. Nash, Jason S. Snyder.

**Formal analysis:** Alyssa M. Ash, Sara Singh, Nicholas P. Vyleta, Jason S. Snyder.

**Funding acquisition:** Jason S. Snyder.

**Investigation:** Alyssa M. Ash, Sue Rim Baek, Peyton Holder, Nicholas P. Vyleta.

**Methodology:** Alyssa M. Ash.

**Resources:** Matthew Willman, Kevin R. Nash.

**Supervision:** Jason S. Snyder.

**Writing – original draft:** Alyssa M. Ash, Jason S. Snyder.

**Writing – review & editing:** Sue Rim Baek, Kevin R. Nash, Jason S. Snyder.

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
