## [Decision Letter · Decision Letter 0]

21 Jul 2025

Dear Dr. Snyder,

Thank you for submitting your manuscript to PLOS ONE. After careful consideration, we feel that it has merit but does not fully meet PLOS ONE’s publication criteria as it currently stands. Therefore, we invite you to submit a revised version of the manuscript that addresses the points raised during the review process.

In the current version of the manuscript reviewers have voiced concerns regarding statistics and interpretation, as well as the way the sex-specific results are presented in addition to minor points. Please address thoroughly the concerns on both reviewers paying particular attention to those of reviewer one who raised some substantial points critical to the conclusions of the manuscript.

We look forward to receiving your revised manuscript.

Kind regards,

Efthimios M. C. Skoulakis, PhD

Academic Editor

PLOS ONE

Journal Requirements:

Reviewers' comments:

Reviewer's Responses to Questions

**Comments to the Author**

1. Is the manuscript technically sound, and do the data support the conclusions?

Reviewer #1: Partly

Reviewer #2: Yes

2. Has the statistical analysis been performed appropriately and rigorously?

Reviewer #1: Yes

Reviewer #2: Yes

3. Have the authors made all data underlying the findings in their manuscript fully available?

Reviewer #1: No

Reviewer #2: Yes

4. Is the manuscript presented in an intelligible fashion and written in standard English?

Reviewer #1: Yes

Reviewer #2: Yes

Reviewer #1: In this manuscript, Ash and co-authors describe a mouse model which better recapitulates sporadic tau pathology in the context of pathological aging and Alzheimer’s disease (AD). They perform various morphometric analysis including dendritic complexity and spine analysis, as well as slice electrophysiology experiments in developmentally- and adult-born dentate gyrus neurons which receive inputs from the lateral entorhinal cortex overexpressing wild-type human 4R tau and GFP, or simply GFP.

The experimental design is neat and targets precisely the neural circuit of interest, which the authors claim to better recapitulate early stages of AD pathology before the spread tau across other brain regions. Generally, the experimental design and data analysis including statistical analysis are robust and convincing. The manuscript is well written and generally clear. I have some concerns regarding the conclusions and interpretation of the results.

Major concerns:

1) The authors found no change in cell density in layer 2 LEC between tau and GFP overexpression using DAPI as a marker. Can they comment on the reason why they used DAPI and not a specific neuronal marker and what the expectation is regarding cell vs neuronal survival in this context?

2) Throughout most of the manuscript the analysis includes sex as a factor in the statistical design for which I commend the authors as we know that there is a substantial difference in the prevalence e.g. of AD between women and men. However, it almost feels like they felt they have to do it rather than actually seriously consider the effect of sex. Information about the impact of sex/gender on tau pathology, aging, AD is missing in the introduction. Moreover, the significant effect of sex on dendritic morphology (Fig.4) is not at all mentioned in the Discussion in the context of the existing literature. No interpretation for molecular mechanism or expected outcomes for either the significant effect on dendritic complexity or lack of significant effect of sex (including main effects and interactions) on the other morphological or electrophysiological measurement.

3) Related to the sex stratified analysis in Fig.4, it is critical to point out that the N=29 and 22 males and 15 and 14 females which represents a substantial difference. Is it possible that the lack of difference between tau and control in females is simply due to lacking enough statistical power? Could the authors provide statistical evidence that that’s not the case and/or discuss the caveat of including twice as many males? The same concern would stand for all cases where there is such a big difference in N (e.g. Fig 5C and E).

4) Is it possible that the lack of significant difference in the LTP experiments (and also in other experiments) stems from inefficient transfection of tau? Can the authors provide some sort of correlation of transfected fibers (intensity or other measure) normalized to surface area/volume/number of L2 LEC cells and EPSC change?

5) The authors interpret the increase of the density of thin spines as well as the increased adult neurogenesis as compensatory neuroplasticity. This is a bit confusing to me, as it has been previously shown that neurogenesis in AD, for example, can be increased or decreased depending on the mouse mode. Furthermore, even though the authors didn’t observe significant changes TBS-induced LTP, between tau and GFP overexpression, the tau potentiation is if anything smaller. The authors should clarify/elaborate this in the Discussion.

Minor concerns:

1) The authors should please refer to all panels of the figures specifically in the text. Not just Figure 3.

2) In some cases (e.g. for Fig.6), not all terms of the statistical model (main effects/interactions) are available in the Results. The authors should please provide them.

3) What statistical tests were used to decompose the interactions? Those should be provided where necessary.

4) The authors do not comment on how injection of tamoxifen itself affects basal neurogenesis, LTP and morphometrics in DG neurons. The experiments are designed such as both groups (control/GFP and tau overexpression both include tamoxifen), which makes the conclusions sound. However, the authors should include references from the literature, so it’s easier for the reader to interpret the data.

5) The statistical results in the result section referring to the data in Fig. 7 A-D are either wrong or misinterpreted. The authors should correct that

“However, for both ABNs and DBNs, there were no statistically significant differences in EPSC size between hTau and GFP mice (Fig. 7A,C; ABNs: effect of stimulation intensity: T221=5.2, P<0.0001; effect of hTau: T14=0.01, P=0.99; effect of sex: T14=0.1, P=0.92; hTau x sex interaction: T14=0.05, P=0.96; ; hTau x stimulation intensity interaction: T221=0.7, P=0.48; hTau x sex x stimulation intensity interaction: T221=3.6, P=0.0004; hTau vs GFP P>0.5 for both males and females; DBNs: effect of stimulation intensity: T230=6.7, P<0.0001; effect of hTau: T13=0.92, P=0.37; effect of sex: T13=0.80, P=0.44; hTau x sex interaction: T13=0.7, P=0.49; hTau x stimulation intensity interaction: T230=3.0, P=0.003; hTau x sex x stimulation intensity interaction: T230=1.9, P=0.06).”

hTau x stimulation intensity interaction: T230=3.0, P=0.003 is completely dismissed and hTau x sex x stimulation intensity interaction: T230=1.9, P=0.06 is similarly dismissed as a trend, whereas similar trend P=0.054 was previously highlighted (“While the hTau x sex interaction was not statistically significant (T24=-2.0, P=0.054), the average dendritic length was 17% lower in male hTau mice as compared to GFP controls, but female hTau mice were 7% higher” )

Reviewer #2: The authors developed a system to model the early-age tau pathology by expressing human tau specifically in LEC brain region. The major phenotype is that mature spines postsynaptically are reduced by hTau, although other aspects including electrophysiology are not changed. I would be intersted in whether the model mouse show cognitive deficits, while that should be beyond the scope of this study.

Minor points:

1. The authors should cite the figures more specifically. For example, they should cite "Figure 4B" instead of "Figure 4", unless they want to refer the whole figure. It occurs for too many times.

2.In Figure 4C, they'd better add x-axis (e.g. distance) with length unit (micrometer).

**Do you want your identity to be public for this peer review?** For information about this choice, including consent withdrawal, please see our Privacy Policy

Reviewer #1: No

Reviewer #2: No

---

## [Author Response · Author response to Decision Letter 1]

4 Sep 2025

Please find below our response to each of the reviewers’ concerns, in bold.

Reviewer #1: In this manuscript, Ash and co-authors describe a mouse model which better recapitulates sporadic tau pathology in the context of pathological aging and Alzheimer’s disease (AD). They perform various morphometric analysis including dendritic complexity and spine analysis, as well as slice electrophysiology experiments in developmentally- and adult-born dentate gyrus neurons which receive inputs from the lateral entorhinal cortex overexpressing wild-type human 4R tau and GFP, or simply GFP.

The experimental design is neat and targets precisely the neural circuit of interest, which the authors claim to better recapitulate early stages of AD pathology before the spread tau across other brain regions. Generally, the experimental design and data analysis including statistical analysis are robust and convincing. The manuscript is well written and generally clear. I have some concerns regarding the conclusions and interpretation of the results.

Major concerns:

1) The authors found no change in cell density in layer 2 LEC between tau and GFP overexpression using DAPI as a marker. Can they comment on the reason why they used DAPI and not a specific neuronal marker and what the expectation is regarding cell vs neuronal survival in this context?

This is a good question. Ideally we certainly would have examined different cell types, not just neurons vs glia but also relevant neuronal subtypes, such as reelin+ cells that project to the DG (the cells we are most interested in targeting). The issue is that our main focus was to verify the AAV injections and resulting Tau/GFP immunostaining. This left us with no tissue to do additional staining for neuronal markers. Since, in humans, vulnerable layer 2 neurons are the major cell types, the absence of any difference suggest that massive neuronal loss was not present in our tissue. We have added text that speaks to this issue in the 1st paragraph of the discussion.

2) Throughout most of the manuscript the analysis includes sex as a factor in the statistical design for which I commend the authors as we know that there is a substantial difference in the prevalence e.g. of AD between women and men. However, it almost feels like they felt they have to do it rather than actually seriously consider the effect of sex. Information about the impact of sex/gender on tau pathology, aging, AD is missing in the introduction. Moreover, the significant effect of sex on dendritic morphology (Fig.4) is not at all mentioned in the Discussion in the context of the existing literature. No interpretation for molecular mechanism or expected outcomes for either the significant effect on dendritic complexity or lack of significant effect of sex (including main effects and interactions) on the other morphological or electrophysiological measurement.

This is a good point. We agree we did not make a sufficient effort to interpret the results in the context of sex differences in AD. Along with some new analyses, in the revised manuscript we have addressed the issue of sex differences more thoroughly in the introduction, results and discussion section.

3) Related to the sex stratified analysis in Fig.4, it is critical to point out that the N=29 and 22 males and 15 and 14 females which represents a substantial difference. Is it possible that the lack of difference between tau and control in females is simply due to lacking enough statistical power? Could the authors provide statistical evidence that that’s not the case and/or discuss the caveat of including twice as many males? The same concern would stand for all cases where there is such a big difference in N (e.g. Fig 5C and E).

We don’t think that the difference in N is leading to a false negative result for dendritic branching in females (Fig 4) because there is not even a trend in the results that would suggest a difference. But we agree that it would be ideal to have similar Ns for males and females. As we have already noted the sample size limitation for the Fig 5 data, we also raise this limitation for the Fig 4 data for consistency.

4) Is it possible that the lack of significant difference in the LTP experiments (and also in other experiments) stems from inefficient transfection of tau? Can the authors provide some sort of correlation of transfected fibers (intensity or other measure) normalized to surface area/volume/number of L2 LEC cells and EPSC change?

This is a good idea but unfortunately we did not keep the slices that were used for recordings and we didn’t systematically keep images from these slices that could be used for any quantitative analyses. What we can say is that all slices were confirmed for GFP expression at the time of recording and so we expect that the quality of expression should be equivalent to what was seen in the tissue that was used for morphological analyses (where we observed some significant effects).

5) The authors interpret the increase of the density of thin spines as well as the increased adult neurogenesis as compensatory neuroplasticity. This is a bit confusing to me, as it has been previously shown that neurogenesis in AD, for example, can be increased or decreased depending on the mouse mode. Furthermore, even though the authors didn’t observe significant changes TBS-induced LTP, between tau and GFP overexpression, the tau potentiation is if anything smaller. The authors should clarify/elaborate this in the Discussion.

Neurogenesis is reduced in the majority of AD mouse models and tau models as well, all of which have much more severe pathology than our model. Since others have found evidence for compensation at early stages (eg with respect to spines and synaptic innervation), we speculate that this may explain our spine and synaptic strength increases. It is indeed possible that it could simply be an effect of the model. In the revised manuscript we raise this alternative interpretation (and also raise the possibility that more spines could be disadvantageous rather than beneficial).

With respect to the LTP experiment, we cleaned up the data by removing any experiments where there was poor sampling of EPSPs during the LTP window (because cells began to spike post-TBS). If anything, there is now greater LTP in the ABN hTau group but, because the effect is non-significant, we are hesitant to discuss it or over interpret it.

Minor concerns:

1) The authors should please refer to all panels of the figures specifically in the text. Not just Figure 3.

This has been done.

2) In some cases (e.g. for Fig.6), not all terms of the statistical model (main effects/interactions) are available in the Results. The authors should please provide them.

These have been added.

3) What statistical tests were used to decompose the interactions? Those should be provided where necessary.

We have clarified this in the methods (in most cases it is Sidak-corrected pairwise comparisons but we also performed simple slopes analyses to look at differences in the ephys input-output curves).

4) The authors do not comment on how injection of tamoxifen itself affects basal neurogenesis, LTP and morphometrics in DG neurons. The experiments are designed such as both groups (control/GFP and tau overexpression both include tamoxifen), which makes the conclusions sound. However, the authors should include references from the literature, so it’s easier for the reader to interpret the data.

We have now added some text to the discussion about how tamoxifen itself could have impacted our results.

5) The statistical results in the result section referring to the data in Fig. 7 A-D are either wrong or misinterpreted. The authors should correct that

“However, for both ABNs and DBNs, there were no statistically significant differences in EPSC size between hTau and GFP mice (Fig. 7A,C; ABNs: effect of stimulation intensity: T221=5.2, P<0.0001; effect of hTau: T14=0.01, P=0.99; effect of sex: T14=0.1, P=0.92; hTau x sex interaction: T14=0.05, P=0.96; ; hTau x stimulation intensity interaction: T221=0.7, P=0.48; hTau x sex x stimulation intensity interaction: T221=3.6, P=0.0004; hTau vs GFP P>0.5 for both males and females; DBNs: effect of stimulation intensity: T230=6.7, P<0.0001; effect of hTau: T13=0.92, P=0.37; effect of sex: T13=0.80, P=0.44; hTau x sex interaction: T13=0.7, P=0.49; hTau x stimulation intensity interaction: T230=3.0, P=0.003; hTau x sex x stimulation intensity interaction: T230=1.9, P=0.06).”

hTau x stimulation intensity interaction: T230=3.0, P=0.003 is completely dismissed and hTau x sex x stimulation intensity interaction: T230=1.9, P=0.06 is similarly dismissed as a trend, whereas similar trend P=0.054 was previously highlighted (“While the hTau x sex interaction was not statistically significant (T24=-2.0, P=0.054), the average dendritic length was 17% lower in male hTau mice as compared to GFP controls, but female hTau mice were 7% higher” )

Thanks for pointing these out. We have gone through these data in depth and found that previous effects and interactions were sometimes interpreted erroneously from the mixed model statistics. We have now analyzed all mixed models using ANOVAs, and report the F statistics and P values from the ANOVA omnibus tables. For the majority of experiments this has not affected the interpretations. However, in the case of these ABN stimulus response data, it revealed a significant hTau x sex x stimulus intensity interaction, which was due to greater hTau effects in males than in females. We have therefore expanded Fig 7 to include breakdowns by sex for the stimulus response data.

With respect to the issue of statistical trends vs effects, we have modified the manuscript and taken the following approach: in instances where an interaction effect was close to the P<0.05 threshold, we now note the near-interaction and follow up with post hoc comparisons but note that these are exploratory analyses that warrant confirmation. We think strikes an appropriate balance between pursuing sex-based analyses while also not overstating possible differences.

Reviewer #2: The authors developed a system to model the early-age tau pathology by expressing human tau specifically in LEC brain region. The major phenotype is that mature spines postsynaptically are reduced by hTau, although other aspects including electrophysiology are not changed. I would be intersted in whether the model mouse show cognitive deficits, while that should be beyond the scope of this study.

We agree if would have been ideal to also examine LEC-based behavioral changes due to hTau. Unfortunately, given limited resources, we were unable to .

Minor points:

1. The authors should cite the figures more specifically. For example, they should cite "Figure 4B" instead of "Figure 4", unless they want to refer the whole figure. It occurs for too many times.

As also suggested by Reviewer 1, this change has been made.

2.In Figure 4C, they'd better add x-axis (e.g. distance) with length unit (micrometer).

Thanks – this has been added.

---

## [Decision Letter · Decision Letter 1]

26 Oct 2025

A mouse model of early sporadic tau pathology induces neurogenic plasticity in the hippocampus

PONE-D-25-18279R1

Dear Dr. Snyder,

We’re pleased to inform you that your manuscript has been judged scientifically suitable for publication by editorial decision since a second reviewer did not send his opinion in a timely manner (apologies for this) and will be formally accepted for publication once it meets all outstanding technical requirements.

Kind regards,

Efthimios M. C. Skoulakis, PhD

Academic Editor

PLOS ONE

Additional Editor Comments (optional):

Reviewers' comments:

Reviewer's Responses to Questions

**Comments to the Author**

Reviewer #1: All comments have been addressed

2. Is the manuscript technically sound, and do the data support the conclusions?

Reviewer #1: Yes

3. Has the statistical analysis been performed appropriately and rigorously?

Reviewer #1: Yes

4. Have the authors made all data underlying the findings in their manuscript fully available?

Reviewer #1: Yes

5. Is the manuscript presented in an intelligible fashion and written in standard English?

Reviewer #1: Yes

Reviewer #1: The authors have addressed the majority of my concerns.

The manuscript now properly discusses the caveats of the experimental design and analyses. The authors have added a section in the discussion on sex differences in their findings and their interpretation in the context of the existing literature, as well as have introduced them properly in the introduction and abstract. The description of the statistical analysis is now clear and the reader can easily follow the experimental design and analysis to be able to draw conclusions from the presented data. The manuscript is valuable description of a new animal model that may capture some cell and circuit properties of early sporadic tau pathology in humans.

Minor comments:

1) In the current version of the manuscript, the figure legend for figure 1 is missing

2) In the figure legend for figure 7, # is marked as P<0.05, I am assuming this is P<0.1? Please correct.

3) The following sentence in the newly added paragraph in the discussion doesn't seem to follow logically from the previous ones, please rewrite: "...In female humans, this might therefore translate into more dramatic or earlier-onset plasticity effects."

**Do you want your identity to be public for this peer review?** For information about this choice, including consent withdrawal, please see our Privacy Policy

Reviewer #1: No

---

## [Editor Report · Acceptance letter]

1 Dec 2025

PONE-D-25-18279R1

PLOS ONE

Dear Dr. Snyder,

I'm pleased to inform you that your manuscript has been deemed suitable for publication in PLOS ONE. Congratulations! Your manuscript is now being handed over to our production team.

Kind regards,

on behalf of

Dr. Efthimios M. C. Skoulakis

Academic Editor

PLOS ONE